# META-COLLABORATION IN DISTILLATION: POOLED LEARNING FROM MULTIPLE STUDENTS

## ABSTRACT

Knowledge distillation (KD) approximates a large *teacher* model using a smaller *student* model. KD can be used to train multiple students of different capacities, allowing for flexible management of inference costs at test time. We propose a novel distillation method we term *meta-collaboration*, wherein a set of students are simultaneously distilled from a single teacher, and can improve each other through information sharing during distillation. We model this information sharing through a separate network designed to predict instance-specific loss mixing for each of the students. This auxiliary network is trained jointly with the multi-student distillation, utilizing a separate meta-loss aggregating student model loss on a separate validation set. Our method improves student accuracy for all students and beats state-of-the-art distillation baselines, including methods that use multi-step distillation, combining models of different sizes. In particular, addition of smaller students to the pool clearly benefits larger student models, through the mechanism of meta-collaboration. We show average gains of 2.5% on CIFAR100 & 2% on TinyImageNet datasets; our gains are consistent across a wide range of student sizes, teacher sizes, and model architectures.

## 1 INTRODUCTION

Practical deployments of machine learning models often need to balance model accuracy against constraints on compute, memory, and inference latency. Recent work in knowledge distillation (KD (Hinton et al., 2015)) offers a helpful tool–train a smaller student model to approximate the higher quality predictions of a larger teacher model. Such distilled student models may perform significantly better than an equivalent model trained from scratch. A known challenge in KD is its poor performance when there is a large capacity gap between the teacher and the student (Cho & Hariharan, 2019). Previous work attempted to bridge this gap by using supervision from multiple models of different sizes, potentially themselves learnt by KD in a sequential manner (Mirzadeh et al., 2020; Son et al., 2021). However often the knowledge transfer is unidirectional *i.e.* from a larger student to a smaller student. In this work, we approach the problem from a complementary perspective: what if multiple students of different sizes could learn from and improve each other through collaboration?

Model collaboration has been leveraged by previous work; *e.g.*, by cross-correlating errors across multiple equivalent models in supervised learning (Baluja et al., 2015), or by creating a pseudo-teacher by pooling multiple peer model predictions in the distillation setting (Chen et al., 2020). Online knowledge distillation, where a strong pre-trained teacher is absent, also utilizes this approach. In Chen et al. (2020), a soft target is obtained by aggregating predictions from peer models of the same architecture, using an attention-based mechanism. In Wu & Gong (2021), the m-branch model treats each branch as a peer model, constructing soft distillation targets from weighted logits of these branches and from an exponentially moving-averaged m-branch model. Additionally, Guo et al. (2020) suggests using pooled student logits with varying learning capacities as soft distillation targets. However, the primary focus in these works lies in constructing logits due to the absence of a pre-trained teacher model. In contrast, our approach centers on influencing the learning patterns of the models participating in the collaborative process. For instance, Du et al. (2023) devise a curriculum of sorts using variance in predictions of multiple students with varying levels of sparsity as a measure of task complexity. Therefore, we aim to harness the advantages of collaborative learning seen in previous research while also benefiting from the introduction of intermediate mod-

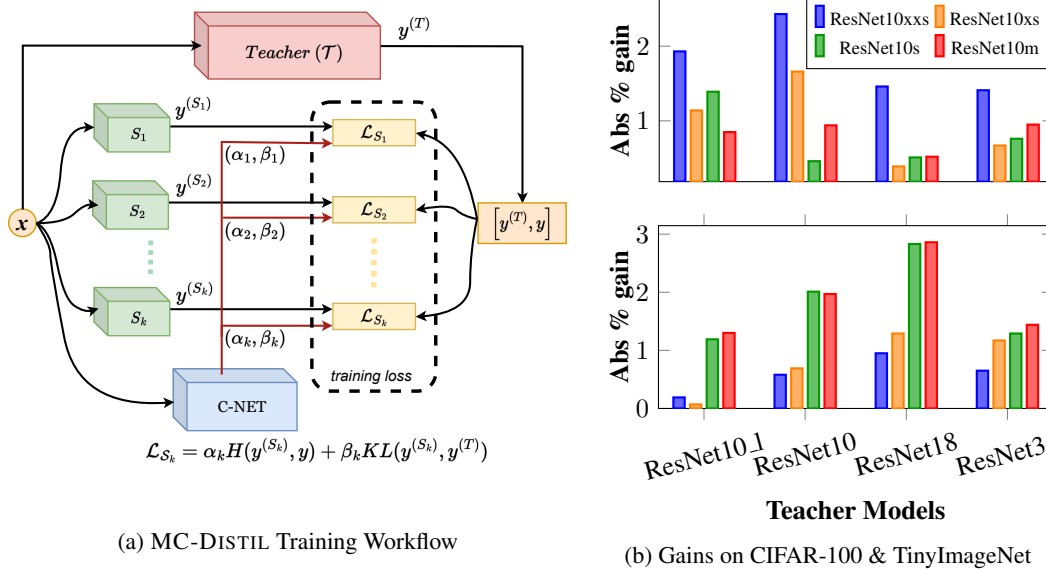

(a) MC-DISTIL Training Workflow

(b) Gains on CIFAR-100 & TinyImageNet

Figure 1: (a) Losses are weighted according to the parameters obtained from C-NET which are then used to train the individual students. (b) Absolute gains over closest baseline for each teacher/student combo, with CIFAR100 (top) and TinyImageNet (bottom) datasets. MC-DISTIL consistently outperforms all baselines across different teachers and student combinations.

els to bridge the gap between teacher and student models. This is not a straightforward task, since techniques such as variance reduction may not be applicable due to the differing learning capacities among the models.

We propose the novel approach of *meta-collaborative distillation* (MC-DISTIL), where students of different capacities collaborate with, and help improve each other, during distillation. The "meta" in meta-collaboration refers to our use of a separate "coordinator" network (C-NET) that synthesizes learnings from the different students, and in turn influences each student's learning process, setting up a meta-learning problem. Our insight is that the C-NET can learn not only about each individual student's performance and needs, but also about training instance characteristics, through the process of evaluating students on input data. As shown in fig. 1a, the C-NET modulates the training loss of each student through an instance-dependent reweighting of the teacher and cross-entropy losses. To set up the C-NET learning objective, we calculate student loss on a separate validation set; this loss is an implicit function of the C-NET through the loss reweighting shown in fig. 1a. In other words, the C-NET learns to weigh student training losses such that their generalization on the validation set is maximized. We develop the modeling objectives and learning algorithm for MC-DISTIL, and evaluate it on a wide range of student & teacher architectures and model sizes, quantifying gains against SOTA benchmarks. MC-DISTIL improves the performance of each student in the ensemble, *highlighting the interesting finding that smaller student models can help improve larger students in a collaborative distillation framework*.

In Figure 1b, we depict the absolute performance improvement achieved by MC-DISTIL compared to the best state-of-the-art (SOTA) baseline when training a group of ResNet (Krizhevsky, 2009) students. Notably, we observe performance enhancements in individual students across various experimental settings. This feature provides our approach with an additional advantage: *it furnishes a spectrum of student models with varying capacities all with improved generalization capabilities*, which can be deployed contextually at test time to suit specific application requirements.

## 2 RELATED WORKS

**Knowledge Distillation (KD)** In supervised learning, Knowledge Distillation (KD) (Hinton et al., 2015) is a valuable method where a 'student' model learns by mimicking a pre-trained 'teacher' model, rather than solely relying on labeled data. The success of KD depends on factors like teacher

model accuracy and student model capacity (Menon et al., 2021). Recent research (Harutyunyan et al., 2023) explores its efficacy in relation to supervision complexity. Using early-stopped teacher models has shown promise in improving student training (Cho & Hariharan, 2019), though it requires iterative distillation. Another approach (Liu et al., 2020) employs multiple teacher networks with intermediate knowledge transfer. Strategic blending of loss components (Sivasubramanian et al., 2023) has recently improved KD, particularly in scenarios with significant representation gaps between teacher and student models. The challenge of suboptimal KD performance caused by significant capacity disparities between student and teacher models was addressed through by strategically choosing points to learn from the teacher model in Kag et al. (2023). Other innovations such as 'Teacher Assistants' (TAs) or intermediate models have been introduced (Mirzadeh et al., 2020), with further enhancements achieved through stochastic techniques such as Dense Gradient Knowledge Distillation (DGKD) (Son et al., 2021), which involves the simultaneous training of intermediates with occasional model dropout. *These works indicate knowledge transfer to the smaller models, resulting in their improvement by virtue of presence of the intermediate bigger models; however the generalization of the larger models has also been shown to improve from the knowledge of smaller models (Mindermann et al., 2022). Therefore, inspired by these bidirectional signals, we present an approach to train multiple student models simultaneously, and communicate vital information via a coordinator network.*

**Instance-Specific Learning:** A substantial body of prior research has delved into instance-specific learning, including the exploration of instance-specific temperature parameters in supervised learning (Saxena et al., 2019). Related literature (Algan & Ulusoy, 2021; Vyas et al., 2020) has also investigated the learning of per-instance *label uncertainty* parameters to account for potential label noise. In the context of knowledge distillation, (Zhao et al., 2021) have demonstrated the advantages of learning instance-level sequences (or curricula) for training samples. A similar instance-wise weighing scheme has been proposed to improve distillation in semi-supervised settings (Iliopoulos et al., 2022). Recent contributions such as those in (Ren et al., 2018; Shu et al., 2019; Raghu et al., 2020) employ meta-learning based on validation sets to acquire instance-specific weights, enhancing robustness. *We introduce a novel approach,* MC-DISTIL, *that involves the utilization of meta-learning based on validation sets to facilitate a collaborative learning process among multiple models. To the best of our knowledge, this has not been previously explored.*

**Bi-level Optimization and Meta-Learning**: In prior research (Jenni & Favaro, 2018; Bengio, 2000; Domke, 2012), there has been an exploration of the learning of network hyper-parameters through the solution of a two-level optimization problem. This entails optimizing on the primary task, and concurrently, on an external meta-task, often involving validation data. These algorithms share similarities with the field of 'learning to learn', which is typically applied in multi-task contexts (Finn et al., 2017; Nichol et al., 2018; Hospedales et al., 2020; Vyas et al., 2020). The conventional approach in these contexts is to acquire a "meta-" algorithm capable of generalizing across tasks by simulating the test dynamics, including the sampling of test tasks, alongside test data, for the assessment and optimization of loss during training (Hospedales et al., 2020). *It is important to note that while both kinds of literature employ nested optimization objectives, our work diverges, in that we are primarily focused on enhancing generalization across different student models via sharing information during joint training using a coordinator network.*

## 3 MC-DISTIL: META-COOPERATIVE DISTILLATION

### 3.1 STANDARD KNOWLEDGE DISTILLATION

Supervised learning trains a classifier $y = f(x)$ using training data $D = (x_i, y_i) \mid i \in (1, \cdots, n)$. Here, $(x_i, y_i) \in \mathcal{X} \times \mathcal{Y}$ denote pairings of inputs $x_i$ and their corresponding labels $y_i$. We parametrize the model $f$ by $\theta \in \Theta$. Typically, the labels $y$ are cardinal in nature, and practitioners have found that such labels are often inadequate in capturing nuances in the data. A popular mitigation is to incorporate more nuanced 'soft labels', or distributions over labels, as the target for supervision instead of cardinal labels. In particular, Knowledge distillation (KD) (Hinton et al., 2015) uses the logits from a pre-trained model (the 'teacher model') as soft labels for training a classifier, in addition to the standard cardinal labels.

Suppose a pre-trained model (teacher) generates logits denoted by $y^{(T)} = \mathcal{T}(\boldsymbol{x})$. Then a new model (student) could be trained using a "teacher matching" objective that involves minimizing the KL-

divergence between $y^{(T)}$ and the student's logits $y^{(S)}$. The KD objective is as follows:

$$\mathcal{L}_s = \sum_D \left( (1-\lambda) l_{ce} + \lambda \left( \tau^2 KL\left(y^{(S)}, y^{(T)}\right) \right) \right) \tag{1}$$

Here, $l_{ce} = H\left(y^{(S)}, y\right)$ is a supervised learning loss matching $y^{(S)}$ to the true labels $y$, and $l_{kd} = \tau^2 KL\left(y^{(S)}, y^{(T)}\right)$ is the teacher-matching loss. Typically, $H$ is the standard cross-entropy loss. The hyperparameters $\tau, \lambda$ control the softening of the KL-divergence term, and the relative contributions of the two loss components.

### 3.2 IMPROVING THE EFFICACY OF KD

Successful application of KD often depends on the quality of the pretrained teacher model (see *e.g.*, (Menon et al., 2021)), and the representational gap between the teacher and student models. For improving KD in general, approaches include (i) early stopping of the teacher model (Cho & Hariharan, 2019), which reduces teacher overfitting, and (ii) instance-based loss mixing (AMAL (Sivasubramanian et al., 2023)). The latter approach uses a training objective specified below:

$$\mathcal{L}_s = \frac{1}{n} \sum_{i=1}^{n} \alpha_i H\left(y_i^{(S)}, y_i\right) + \beta_i \tau^2 KL\left(y_i^{(S)}, y_i^{(T)}\right) \tag{2}$$

Here, $\alpha_i, \beta_i$ are weights that control the contributions of the KD loss components at an instance level. These parameters improve knowledge transfer even in settings with a large capacity gap between the teacher and student models, although at the cost of learning a large number of free parameters $(\alpha_i, \beta_i)$ equal to the size of the training dataset.

Other approaches have addressed the capacity gap directly, for instance by using Teacher Assistants (TA) or intermediate models (Mirzadeh et al., 2020). Teacher knowledge is first distilled to an intermediate-sized model, which is then used to teach the student model. This idea was then extended to multi-step sequential distillation where the sequence of KD steps may depend on the size of the capacity gap. Other work sped up this inherently sequential process by using an ensemble of differently-sized teachers (DGKD (Son et al., 2021)) in a stochastic manner with occasional teacher model dropout.

### 3.3 MC-DISTIL: MULTI STUDENT KNOWLEDGE DISTILLATION

The success of the multi-teacher approaches described above (Mirzadeh et al., 2020; Son et al., 2021) suggests that supervisory inputs from teacher models of different sizes enrich the information available to the student. We leverage this insight in a completely different setup where a single teacher is simultaneously distilled into multiple cooperating student models. Our primary thesis is that by integrating the performance characteristics of different students (and the teacher) on a training instance, we can learn how to effectively customize the distillation loss for each student.

Formally, we assume a series of student models $\mathcal{S} = \{S_j | j \in \{1, \cdots, k\}\}$, and a single pre-trained teacher model $\mathcal{T}$. We start with the MC-DISTIL training objective in its basic form:

$$\mathcal{L}_{s_j} = \frac{1}{n} \sum_{i=1}^{n} \alpha_{ij} H\left(y_i^{(S_j)}, y_i\right) + \beta_{ij} \tau^2 KL\left(y_i^{(S_j)}, y_i^{(T)}\right) \qquad \forall j \in \{1, \cdots, k\} \tag{3}$$

*i.e.*, each student has its own set of instance-specific loss mixing parameters, similar to eq. (2).

**Coordinator network:** Recall that in AMAL (eq. (2)), the loss-mixing weights are entirely free parameters, leading to a large number of learnable parameters. Further, the student losses as written in eq. (3) do not interact with each other as yet. Our key innovation is the use of an additional "coordinator network", that we call C-NET, to address both these challenges (fig. 1a). The C-NET is a learnt function $(A, B) = g_\phi(x)$, where $A = [\alpha_1, \ldots, \alpha_k], B = [\beta_1, \ldots, \beta_k]$ represent the loss mixing parameters for the $k$ students on the input $x$. In this manner, not only are the loss mixing parameters compactly represented by the C-NET parameters $\phi \in \Phi$, but also all learning across students and training instances is channeled through the single point of the coordinator network. The training procedure for the C-NET is described in the following section.

**Consensus across students:** To further encourage information sharing across students, we propose an additional term in the loss function eq. (3). This term minimizes the KL divergence between each student's logits and a *consensus* representation of logits across students (a representation we call the "Pooled Student"). For a training instance $(x, y) \in D$ such that the true label is $c$, *i.e.*, $y[c] = 1$, we first define the PooledStudent logits for each label $l$:

$$y^{(PS)}[l] = \begin{cases} \max\left(y^{(S_1)}[l], \cdots y^{(S_k)}[l]\right), \text{ if } l = c \\ \min\left(y^{(S_1)}[l], \cdots y^{(S_k)}[l]\right), \text{ otherwise} \end{cases} \tag{4}$$

This is similar to MinLogit introduced in (Guo et al., 2020). Using the PooledStudent logits, the loss function in eq. (3) is updated to:

$$\mathcal{L}_{s_j} = \frac{1}{n} \sum_{i=1}^{n} \alpha_{ij} H\left(y_i^{(S_j)}, y_i\right) + \beta_{ij}\tau^2 KL\left(y_i^{(S_j)}, y_i^{(T)}\right)$$

$$+ \gamma_{ij}\tau^2 KL\left(y_i^{(S_j)}, y_i^{(PS)}\right) \qquad \forall j \in \{1, \cdots, k\} \tag{5}$$

Correspondingly, the C-NET outputs are extended to be $(A, B, \Gamma) = g_\phi(x)$. Putting it all together, MC-DISTIL minimizes the total loss over all students, *i.e.*, $\mathcal{L}_s = \sum_j \mathcal{L}_{s_j}$

## 3.4 Training the C-Net

We train the C-NET $g_\phi$ using a separate validation set of data $\mathcal{V} = (x_i^v, y_i^v) \mid i \in (1, \cdots, m)$. The model $g_\phi$ is learnt so as to minimize the average of the student losses on the validation data, *i.e.*,

$$\mathcal{L}_{\text{C-NET}} = \sum_{j=1}^{k}\sum_{i=1}^{m} H\left(y_i^{v(S_j)}, y_i^v\right) = \sum_{j=1}^{k}\sum_{i=1}^{m} H\left(f_{\theta_j}(x_i^v), y_i^v\right) \tag{6}$$

Here $H$ is chosen to be cross-entropy, and summed over students $f_{\theta_j}(\cdot)$ and validation instances. The meta-loss $\mathcal{L}_{\text{C-NET}}$ depends on $\phi$ indirectly through the student model parameters $\theta_j$, which are themselves optimized using the outputs of $g_\phi(\cdot)$ (eq. (4)). In other words, our proposal defines a *bi-level optimization* that encompasses both the C-NET and classifier parameters. This is due to the mutual influence between the optimization objectives of each parameter set, $\theta$ and $\phi$. Formally:

$$\theta_j^* = \operatorname*{argmin}_{\theta} \frac{1}{n}\sum_{i=1}^{n} A[j] \cdot H\left(y_i^{(S_j)}, y_i\right) + B[j] \cdot \tau^2 KL\left(y_i^{(S_j)}, y_i^{(T)}\right)$$

$$+ \Gamma[j] \cdot \tau^2 KL\left(y_i^{(S_j)}, y_i^{(PS)}\right) \qquad \forall j \in \{1, \cdots, k\}$$

$$\text{s.t. } \phi^* = \operatorname*{argmin}_{\phi} \mathcal{L}_{\text{C-NET}}(\mathcal{V}, \theta_1, \cdots, \theta_k) \tag{7}$$

where $(A, B, \Gamma) = g_{\phi^*}(\cdot)$ are loss-mixing weights output using the optimal C-NET parameters $\phi^*$. From the equations above, clearly the optimal $\theta$ values depend on the optimal choice of $\phi$. Equally, to obtain the optimal $\phi$ values, one needs optimal $\theta$ values, since $\mathcal{L}_{\text{C-NET}}$ uses the student models $f_{\theta_1}, \ldots, f_{\theta_k}$. Instead of completely solving the inner loop (optimizing $\phi$) for every setting of the outer parameters $\theta$, we use alternating stochastic gradient descent to devise a tractable learning algorithm. The updates can be summarized as follows:

$$\theta_j^{t+1} = \theta_j^t - \frac{\eta_1^j}{n}\sum_{i=1}^{n} g_{\phi^t}^\alpha[j] * \nabla_{\theta_j^t} H\left(y_i^{(S_j)}, y_i\right)$$

$$+ g_{\phi^t}^\beta[j] * \tau^2 * \nabla_{\theta_j^t} KL\left(y_i^{(S_j)}, y_i^{(T)}\right)$$

$$+ g_{\phi^t}^\gamma[j] * \tau^2 * \nabla_{\theta_j^t} KL\left(y_i^{(S_j)}, y_i^{(PS)}\right) \quad \forall j \in \{1, \cdots, k\} \tag{8}$$

$$\phi^{t+1} = \phi^t - \frac{\eta_2}{m}\sum_{i=1}^{m} \nabla_{\phi^t} \mathcal{L}_{\text{C-NET}}(x_i^v, y_i^v, \theta_1^{t+1}, \cdots, \theta_k^{t+1})$$

$$= \phi^t - \frac{\eta_2}{m}\sum_{j=1}^{k}\sum_{i=1}^{m} \nabla_{\theta_j^{t+1}} \mathcal{L}_{\text{C-NET}}(x_i^v, y_i^v, \theta_1^{t+1}, \cdots, \theta_k^{t+1}) * \nabla_{\phi^t}\theta_j^{t+1} \tag{9}$$

---

**Algorithm 1** The MC-DISTIL approach: learning student $\mathcal{S}_1, \cdots \mathcal{S}_k$, Training data $\mathcal{D}$, Validation data $\mathcal{V}$, teacher $\mathcal{T}$ and C-NET $g_\phi$.

---

**Hyperparameters:** $\tau$ Temperature, $\eta_1^1 \cdots \eta_1^k$: learning rates for $k$ students, $\eta_2$ learning rate for C-NET

---

1: Initialize student model parameters with $\theta^{(0)} \cdots \theta^{(k)}$ and C-NET with $g_\phi^{(0)}$
2: **for** $t \in \{0, \ldots, T\}$ **do**
3:     Update $\theta^{t+1}$ by Equation 8.
4:     **if** t % L == 0 **then**
5:         $\{x^v, y^v\} \leftarrow$ SampleMiniBatch($\mathcal{V}$)
6:         Compute $\mathcal{L}_{\text{C-NET}}$ using $\{x^v, y^v\}$ as described in Equation 6 using the recently updated parameters.
7:         Update $\phi^{\lfloor \frac{t}{L} \rfloor + 1}$ by Eq. Equation 9.
8:     **end if**
9: **end for**

---

Here $\eta_1^1, \cdots, \eta_1^k$ and $\eta_2$ are learning rates corresponding to the various student model training and the C-NET training respectively. The update step for the C-NET is similar to the standard meta-learning objectives as it uses the updated student model parameters. We present the complete algorithm of MC-DISTIL in Algorithm 1. Since, training C-NET adds to the cost of training, we propose to update C-NET only after $L$ epochs.

## 4 EXPERIMENTS

### 4.1 MODEL ARCHITECTURE AND TRAINING

To demonstrate the utility of our method across groups of different model sizes we experiment with a group of ResNet (He et al., 2016) models and several recent larger models. We use the ResNet32 model as C-NET with the classification head changed to output weighting parameters. We use ResNet10-xxxs, ResNet10-xxs, ResNet10-xs, ResNet10-s and ResNet10-m (Kag et al., 2023) models as the student models. These models are simultaneously trained with either ResNet-10L, ResNet-10, ResNet-18 or ResNet-34 models as a teacher model. In Section A.2 we present details of these models. We present experiment results on these combinations in Table 1. To illustrate the utility of our method in larger vision models we perform knowledge distillation with ResNet-32x4 as the teacher and ResNet-8x4, ShuffleNet-V2 (Ma et al., 2018), WideResNet-16x2 (Zagoruyko & Komodakis, 2016) and MobileNet-V2x2 (Sandler et al., 2018) as the group of student models. We also perform knowledge distillation with WideResNet-40x2 as a teacher model and ResNet-8x4, ShuffleNet-V2, WideResNet-40x1 and MobileNet-V2x2 as a student model group in the large vision model setting. The results of these experiments are presented in Table 2.

We train the student models for 500 epochs and update C-NET every 20 epoch (*i.e.*, L = 20). Other training related details are presented in Section A.3. We perform all our experiments on the **CIFAR-100** (Krizhevsky, 2009) and **Tiny-ImageNet** (Le & Yang, 2015) datasets. Details of train-val-test splits, input dimensions, and the augmentations used on the input to model are presented in Appendix A.1.

### 4.2 BASELINES

In our comparative analysis, we assess the performance of our method against a selection of recent works in the field of knowledge distillation, with a specific emphasis on scenarios involving multiple students or intermediate models. This evaluation is conducted alongside standard knowledge distillation and Empirical Risk Minimization (ERM). We specifically highlight three notable recent publications, chosen as representative benchmarks:

**Teacher Assistant Knowledge Distillation (TAKD)** (Mirzadeh et al., 2020) This approach introduces a multi-step distillation process that leverages intermediate-level teachers to facilitate the efficient transfer of knowledge from a large pretrained teacher network to a more compact student

Table 1: Comprehensive comparison of methods across datasets. Columns represent various baselines, alongside the Teacher model and MC-DISTIL. For each teacher model, we perform knowledge distillation with a group of student models. Rows show average accuracy on unseen test data. MC-DISTIL substantially improves the test accuracy compared to other distillation baselines especially the ones designed to take advantage of a multi-student setup. The highest accuracies are highlighted in bold.

| CIFAR100 Test Accuracies | | | | | | | | |
|---|---|---|---|---|---|---|---|---|
| **Teacher** | | **Student** | **CE** | **KD** | **TAKD** | **DGKD** | **RMC** | **Meta-Distill** | **MC-DISTIL** |
| ResNet10-l | 72.2 | ResNet10-xxs | 31.85 | 33.45 | 34.39 | 35.34 | 34.07 | 34.92 | **37.27** |
| | | ResNet10-xs | 42.75 | 44.87 | 44.97 | 47.11 | 45.18 | 46.01 | **48.25** |
| | | ResNet10-s | 52.48 | 55.38 | 56.16 | 57.02 | 53.74 | 57.2 | **58.59** |
| | | ResNet10-m | 64.28 | 66.93 | 67.12 | 67.4 | 66.66 | 68.28 | **69.13** |
| ResNet10 | 75.18 | ResNet10-xxs | 31.85 | 33.95 | 34.98 | 34.85 | 33.64 | 34.66 | **37.28** |
| | | ResNet10-xs | 42.75 | 44.87 | 45.64 | 46.68 | 42.45 | 46.32 | **48.34** |
| | | ResNet10-s | 52.48 | 55.56 | 56.51 | 56.84 | 53.64 | 57.78 | **58.24** |
| | | ResNet10-m | 64.28 | 67.27 | 67.82 | 67.94 | 66.58 | 68.89 | **69.83** |
| ResNet18 | 76.99 | ResNet10-xxs | 31.85 | 33.56 | 34.26 | 34.26 | 33.77 | 34.4 | **36.22** |
| | | ResNet10-xs | 42.75 | 45.02 | 45.27 | 47.33 | 45.14 | 46.24 | **47.72** |
| | | ResNet10-s | 52.48 | 55.73 | 55.41 | 56.7 | 54. 03 | 57.4 | **57.91** |
| | | ResNet10-m | 64.28 | 66.42 | 66.04 | 67.35 | 66.04 | 68.43 | **68.95** |
| ResNet34 | 79.47 | ResNet10-xxs | 31.85 | 33.32 | 34.46 | 35.64 | 34.46 | 33.76 | **37.05** |
| | | ResNet10-xs | 42.75 | 44.94 | 45.92 | 47.21 | 42.78 | 46.43 | **47.88** |
| | | ResNet10-s | 52.48 | 54.73 | 56.17 | 57.12 | 53.58 | 56.91 | **57.88** |
| | | ResNet10-m | 64.28 | 66.52 | 67.47 | 67.55 | 65.58 | 68.09 | **69.04** |
| **Tiny-ImageNet Test Accuracies** | | | | | | | | |
| ResNet10-l | 41.25 | ResNet10-xxs | 13.76 | 13.53 | 13.81 | 14.34 | 13.69 | 14.78 | **14.97** |
| | | ResNet10-xs | 18.56 | 19.19 | 19.22 | 20.54 | 19.04 | 20.13 | **20.61** |
| | | ResNet10-s | 24.56 | 25.95 | 26.35 | 27.24 | 25.86 | 27.06 | **28.24** |
| | | ResNet10-m | 33.47 | 34.63 | 34.86 | 34.61 | 33.72 | 36.02 | **37.32** |
| ResNet10 | 44.04 | ResNet10-xxs | 13.76 | 13.8 | 14.01 | 14.52 | 13.83 | 14.19 | **15.1** |
| | | ResNet10-xs | 18.56 | 19.48 | 19.09 | 21.21 | 19.28 | 20.13 | **21.9** |
| | | ResNet10-s | 24.56 | 26.95 | 25.58 | 26.99 | 26.18 | 27.06 | **29.11** |
| | | ResNet10-m | 33.47 | 35.5 | 35.03 | 35.28 | 34.78 | 36.02 | **38.27** |
| ResNet18 | 47.94 | ResNet10-xxs | 13.76 | 14.12 | 14.53 | 13.87 | 14.08 | 14.24 | **15.19** |
| | | ResNet10-xs | 18.56 | 19.78 | 19.35 | 19.54 | 19.75 | 19.96 | **21.25** |
| | | ResNet10-s | 24.56 | 26.3 | 26.17 | 27.42 | 25.08 | 27.32 | **30.25** |
| | | ResNet10-m | 33.47 | 35.08 | 35.02 | 35.28 | 33.37 | 36.08 | **38.94** |
| ResNet34 | 50.1 | ResNet10-xxs | 13.76 | 14.43 | 13.47 | 14.58 | 13.78 | 13.96 | **15.23** |
| | | ResNet10-xs | 18.56 | 19.72 | 18.33 | 20.84 | 19.28 | 20.93 | **22.1** |
| | | ResNet10-s | 24.56 | 27.05 | 24.96 | 27.89 | 25.99 | 27.64 | **29.18** |
| | | ResNet10-m | 33.47 | 35.94 | 35.94 | 35.6 | 33.58 | 36.88 | **38.32** |

model. To realize this, we employ a teacher network identical to our own and enlist fellow student models with higher learning capacities to serve as intermediate models in this knowledge distillation process.

**Densely Guided Knowledge Distillation (DGKD)**(Son et al., 2021) Much like TAKD, this approach employs several intermediate models; however, it distinguishes itself by training the final student model through a single distillation step. In addition to the teacher KL divergence loss, the training objective for the final student model incorporates the KL divergence loss obtained from the pretrained intermediate models.

**Robust Model Compression (RMC)**(Du et al., 2023). It uses multiple students with various levels of sparsity and interprets the variance in their predictions for each instance as a measure of task complexity. Subsequently, it refines the teacher predictions based on this complexity metric, re-

Table 2: Comprehensive comparison of methods when training models with larger learning capacity. Here again, columns represent baselines, Teacher model accuracy and MC-DISTILRows show average accuracy on unseen test data. MC-DISTIL substantially improves the test accuracy compared to other distillation baselines, especially the ones designed to take advantage of a multi-student setup. The highest accuracies are highlighted in bold.

| CIFAR100 Test Accuracies | | | | | | | | |
|---|---|---|---|---|---|---|---|---|
| **Teacher** | | **Student** | **CE** | **KD** | **TAKD** | **DGKD** | **RMC** | **Meta-Distill** | **MC-DISTIL** |
| ResNet-32x4 | 80.12 | ResNet-8x4 | 71.12 | 72.62 | 74.26 | 74.45 | 73.89 | 73.12 | **75.38** |
| | | ShuffleNet-V2 | 73.73 | 75.33 | 76.89 | 77.24 | 76.52 | 76.74 | **77.97** |
| | | WideResNet-16x2 | 72.79 | 73.34 | 73.72 | 74.12 | 75.19 | 74.1 | **75.52** |
| | | MobileNet-V2x2 | 69.52 | 71.78 | 72.07 | 72.27 | 71.23 | 72.53 | **73.05** |
| WideResNet-40x2 | 77.67 | ResNet-8x4 | 71.12 | 72.77 | 73.82 | 74.63 | 73.84 | 74.49 | **76.73** |
| | | ShuffleNet-V2 | 73.73 | 75.85 | 77.59 | 78.05 | 76.72 | **78.17** | 77.94 |
| | | WideResNet-40x1 | 72.90 | 73.01 | 73.72 | 74.67 | 75.44 | 75.38 | **75.85** |
| | | MobileNet-V2x2 | 69.52 | 72.69 | 72.69 | 71.61 | 70.90 | 74.20 | **74.36** |
| Tiny-ImageNet Test Accuracies | | | | | | | | |
| ResNet-32x4 | 50.24 | ResNet-8x4 | 37.16 | 37.23 | 38.83 | 39.52 | 36.76 | 40.46 | **41.89** |
| | | ShuffleNet-V2 | 47.76 | 50.44 | 49.80 | 50.40 | 49.46 | 50.62 | **52.38** |
| | | WideResNet-16x2 | 39.11 | 39.47 | 41.66 | 41.77 | 39.77 | 42.12 | **43.88** |
| | | MobileNet-V2x2 | 47.68 | 49.89 | 49.89 | 48.21 | 48.07 | 49.85 | **49.95** |

sulting in a more robust knowledge distillation process. In our experiments, we employ students with diverse learning capacities as a substitute for models with different levels of sparsity, achieving similar benefits.

We also compare MC-DISTIL against one more baseline that involves distilling knowledge to each of the students independently using a network architecturally similar to C-NET. We refer to this baseline as the **Meta-Distil**. This baseline is similar to AMAL (Sivasubramanian et al., 2023); the strategic mixing loss components are achieved via the C-NET optimization.

## 4.3 IMPROVING EFFICACY OF KNOWLEDGE DISTILLATION

In Table 1, we present results from experiments conducted on CIFAR100 and TinyImagenet datasets, exploring scenarios with a significant capacity gap between teacher and student models. We start with ResNet10-l as the teacher and go on continuing the increasing learning capacity of the teacher model and perform knowledge distillation with ResNet10, ResNet18 and ResNet34. On both the datasets, MC-DISTIL, by virtue of meta-collaboration, achieves the best performance among the baselines showing accuracy gains of up to 4% on both the datasets compared to KD. The gains are much more pronounced on the larger models in the student pool for the TinyImagenet dataset owing to the increased difficulty in classifying it whereas for CIFAR100 the gains are pretty uniform across the student models. These gains are consistent across a wide range of student and teacher capacities. MC-DISTIL improves all of the student model's performances as compared to the baselines, thereby showing that joint distillation of knowledge to a student set is beneficial for both smaller and larger students. We perform all our experiments with the loss described in equation 5 *i.e.* with PooledStudent logits. We present an ablation on the effect of model performance when models are trained with PooledStudent logits in Appendix B.1.

MC-DISTIL *remains competitive even in scenarios with a small capacity difference.* As illustrated in Table 2, MC-DISTIL maintains its competitive advantage over KD, even when the student model closely matches the size of the teachers, achieving gains of up to 3% relative to KD. These improvements can be attributed to two key factors: (i) the reweighing of loss terms and (ii) meta-collaboration. The reweighing strategy effectively imposes a curriculum, prioritizing the learning of easier points according to the model's capacity, which elucidates the performance gain observed with Meta-Distill compared to KD for students such as 'ResNet-m'. However, just this reweighing is not sufficient for students such as 'ResNet-xxs' and 'Resnet-s' in case of larger teachers. This is where MC-DISTIL's ability to leverage C-NET as a communication channel among student models is useful in enhancing knowledge transfer from the teacher model. While the benefits of information flow from intermediate models to smaller ones to improve final performance, as demonstrated in

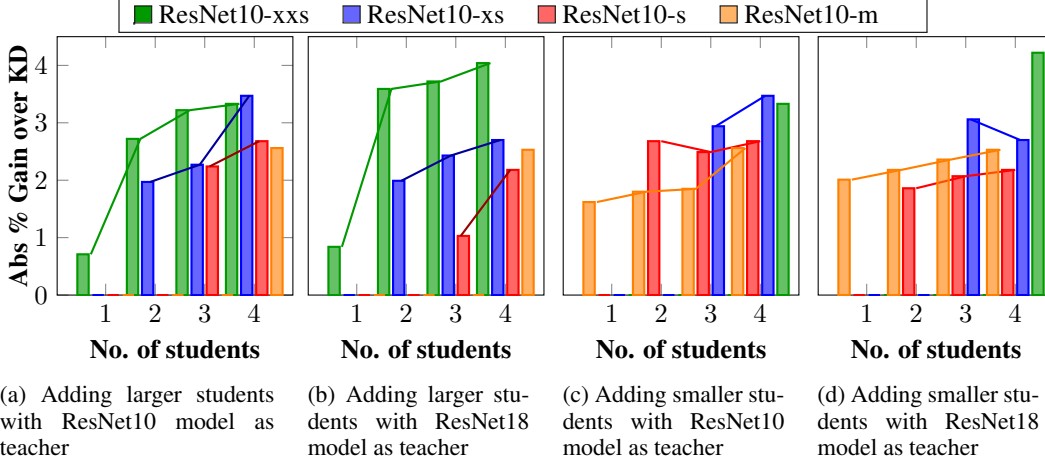

(a) Adding larger students with ResNet10 model as teacher

(b) Adding larger students with ResNet18 model as teacher

(c) Adding smaller students with ResNet10 model as teacher

(d) Adding smaller students with ResNet18 model as teacher

Figure 2: Analysis of the effect of introducing student model cohorts for different learning capacities in meta-collaboration setting.

previous studies (Son et al., 2021; Mirzadeh et al., 2020), are well-established, the reverse scenario has been under-explored. The gains reported in Tables 1 and 2 clearly indicate that larger models can also benefit from information exchange with smaller models, akin to standard supervised settings (Mindermann et al., 2022).

MC-DISTIL *outperforms KD when using less capable teachers compared to KD's best teachers.* As shown in Table 1, KD exhibits an increase in student performance as teacher complexity rises for a given student model. However, it is worth noting that MC-DISTIL achieves significantly superior performance with even less capable teachers, exemplified by the 'ResNet10-s' student. When trained on the TinyImagenet dataset, KD boosts accuracy from 25.95% to 27.05% with high-capacity teachers, whereas 'ResNet10-s' trained with MC-DISTIL achieves an impressive accuracy of 28.24% even when taught by the least capable teacher, 'ResNet10-l'. This not only leads to improved performance but also conserves valuable resources, as it obviates the need for large, computationally intensive teacher models. Instead, our approach advocates for the use of a cost-effective smaller student model to enhance final model performance.

### 4.4 ABLATIONS: CHANGING NO. OF STUDENT COHORTS

To investigate the impact of introducing additional students in the presence of C-NET, we conducted an experiment in which we incrementally introduced student models, one at a time. We present the results of these experiments in Figure 2. This experiment was conducted in two distinct settings: one in which each subsequent addition involved a student with a larger learning capacity. This is presented in Figure 2a and 2b. The other setting is in which each additional introduced student is of a smaller learning capacity as shown in Figure 2c and 2d. In both the settings, we perform knowledge distillation with two teachers *viz.,* ResNet10 and Resnet18. We note that across different teachers, introducing additional students improves the performances of all the students participating in the training process. This is much more pronounced in the setting where larger students are added.

## 5 CONCLUSION

In this paper, we introduce MC-DISTIL, a novel knowledge distillation framework based on meta-collaboration. Through collaborative learning among a group of student models facilitated by a co-ordinating network C-NET, we enhance each individual student model's performance. Our student pool includes models with varying learning capacities. Thus, in addition to leveraging teacher model signals, our approach taps into insights from peer student models to improve individual models' performance. We validate these claims through extensive experiments with various teacher-student combinations and datasets. Our results consistently demonstrate MC-DISTIL outperforming several state-of-the-art knowledge distillation methods involving intermediate models

## REPRODUCIBILITY STATEMENT

In Section 4.1, we introduce several smaller ResNet models, specifically ResNet10-xxxs, ResNet10-xxs, ResNet10-xs, ResNet10-s, ResNet10-m, and ResNet10-l. These models are adopted from Kag et al. (2023), and in Table 3, we compare their architectural details with well-established ResNet models like ResNet10, ResNet18, and ResNet34. Similar to the standard ResNet models, these newer, more compact versions also employ the traditional 'BasicBlock' as their fundamental building block. The architectural structure consists of a convolutional block, four stages of residual blocks, an adaptive average pooling layer, a convolutional block, and a classifier layer. The sole variation among the various capacity variants within this experimental setup is the number of filters in each stage and the residual block. Additionally, Table 3 provides information about the number of parameters and multiply-addition (MAC) operations for each model for the datasets CIFAR-100 and Tiny-ImageNet.

For all the KD baselines listed in Section 4.2 we use temperature $\tau = 2$. We employ the SGD optimizer to train the student models and ADAM optimizer (Kingma & Ba, 2014) to train C-NET. We use a batch size of 400 for both the CIFAR100 and TinyImagenet datasets, We train the student models for 500 epochs and update C-NET every 20 epoch (*i.e.*, L = 20). We use cosine annealing (Loshchilov & Hutter, 2017) as the learning rate schedule for training the student models. We warm start each student model by first training it using the cross entropy loss without using the teacher model for all KD baselines. For both the datasets, we use a learning rate of $0.05$, set weight decay to $1e - 4$ and momentum to $0.95$. For the C-NET training we use a learning rate of $1e - 3$ and set weight decay to $1e - 4$.

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

# Appendix

## A  TRAINING DETAILS

### A.1  DATASET DETAILS

We conduct experiments using the following real-world datasets to showcase the effectiveness of our approach,

**CIFAR-100** (Krizhevsky, 2009). The dataset consists of a total of 60K examples, distributed across 100 distinct classes. Each example in this dataset comprises images with a resolution of $32 \times 32 \times 3$. Specifically, the training set encompasses 50,000 examples, while the remaining 10K serve as the testing set. In our experimental setup, approximately 5K examples are allocated for use as a validation set for our C-NET. For the other baseline models, these validation examples are utilized for the purpose of hyper-parameter tuning.

**Tiny-ImageNet** (Le & Yang, 2015). This dataset is derived from the extensive ImageNet-1K dataset (Russakovsky et al., 2015). This dataset encompasses a total of 100K images, which have been downsampled to a $64 \times 64$ resolution, representing a subset of 200 classes mirroring those in the ImageNet-1K dataset, with each class containing precisely 500 images. As part of our experimental protocol, we have set aside an independent validation set comprising 10K examples, which is utilized for both our proposed method and the baseline models.

For both the datasets, we use the data augmentations methods, using the torchvision's transforms module. We use RandomCrop, RandomResizedCrop, RandomSizedCrop, RandomHorizontalFlip, Normalize, ColorJitter for our purpose. Augmentation methods applied on both dataset, over all experiments, baselines, over all models.

### A.2  MODEL DETAILS

In Section 4.1, we introduce several smaller ResNet models, specifically ResNet10-xxxs, ResNet10-xxs, ResNet10-xs, ResNet10-s, ResNet10-m, and ResNet10-l. These models are adopted from Kag et al. (2023), and in Table 3, we compare their architectural details with well-established ResNet models like ResNet10, ResNet18, and ResNet34. Similar to the standard ResNet models, these newer, more compact versions also employ the traditional 'BasicBlock' as their fundamental building block. The architectural structure consists of a convolutional block, four stages of residual blocks, an adaptive average pooling layer, a convolutional block, and a classifier layer. The sole variation among the various capacity variants within this experimental setup is the number of filters in each stage and the residual block. Additionally, Table 3 provides information about the number of parameters and multiply-addition (MAC) operations for each model for the datasets CIFAR-100 and Tiny-ImageNet.

| Architecture | Filters | Basic block Repeats | CIFAR-100 | | Tiny-Imagenet | |
|---|---|---|---|---|---|---|
| | | | MACs | Params | MACs | Params |
| ResNet10-xxs | [8, 8, 16, 16] | [1, 1, 1, 1] | 2 M | 13 K | 8 M | 15 K |
| ResNet10-xs | [8, 16, 16, 32] | [1, 1, 1, 1] | 3 M | 28 K | 12 M | 31 K |
| ResNet10-s | [8, 16, 32, 64] | [1, 1, 1, 1] | 4 M | 84 K | 16 M | 90 K |
| ResNet10-m | [16, 32, 64, 128] | [1, 1, 1, 1] | 16 M | 320 K | 64 M | 333 K |
| ResNet10-l | [32, 64, 128, 256] | [1, 1, 1, 1] | 64 M | 1.25 M | 255 M | 1.28 M |
| ResNet10 | [64, 128, 256, 512] | [1, 1, 1, 1] | 253 M | 4.92 M | 1013 M | 5 M |
| ResNet18 | [64, 128, 256, 512] | [2, 2, 2, 2] | 555 M | 11.22 M | 2221 M | 11.27 M |
| ResNet34 | [64, 128, 256, 512] | [3, 4, 6, 3] | 1159 M | 21.32 M | 4637 M | 21.38 M |

Table 3: Comparision of newly introduced smaller ResNet with the standard ResNet models

### A.3 HYPER-PARAMETERS

For all the KD baselines listed in Section 4.2 we use temperature $\tau = 2$. We employ the SGD optimizer to train the student models and ADAM optimizer (Kingma & Ba, 2014) to train C-NET. We use a batch size of 400 for both the CIFAR100 and TinyImagenet datasets, We train the student models for 500 epochs and update C-NET every 20 epoch (*i.e.*, L = 20). We use cosine annealing (Loshchilov & Hutter, 2017) as the learning rate schedule for training the student models. We warm start each student model by first training it using the cross entropy loss without using the teacher model for all KD baselines. For both the datasets, we use a learning rate of $0.05$, set weight decay to $1e - 4$ and momentum to $0.95$. For the C-NET training we use a learning rate of $1e - 3$ and set weight decay to $1e - 4$.

## B ADDITIONAL EXPERIMENTS

We have released anonymized code at the URL: `https://anonymous.4open.science/r/Multinet-E01D`.

### B.1 EFFECT ADDING POOLED STUDENT

| Teacher | Student | MC-DISTIL-NoPS | MC-DISTIL |
|---|---|---|---|
| ResNet10-l | ResNet10-xxs | 36.29 | **36.44** |
| | ResNet10-xs | 47.14 | **47.65** |
| | ResNet10-s | 57.37 | **57.55** |
| | ResNet10-m | 68.47 | **68.51** |
| ResNet10 | ResNet10-xxs | 36.57 | **36.78** |
| | ResNet10-xs | 47.55 | **48.03** |
| | ResNet10-s | 57.59 | **58.2** |
| | ResNet10-m | 69.03 | **69.5** |

Table 4: We present the analysis of the effect of adding loss component based on the pooled student to MC-DISTIL. Here MC-DISTIL-NoPS represents the test result obtained with model trained without pooled student based loss component.

We analyze the effect of adding an additional loss component associated with a fake teacher introduced in equation 4 in Section 3.3. The PooledStudent logit is composed of logits from fellow student models and was introduced to improve communication amongst the student models. We present test accuracy obtained on CIFAR100 dataset in Table 4 where MC-DISTIL-NoPS represents training without addition of loss component associated with a fake teacher. We note that training with this new logit is helpful to boost the performance gains obtained via using meta-collaborative learning.

