# OpenReview forum: "Meta-Collaboration in Distillation: Pooled Learning from Multiple Students"
_ICLR.cc/2024/Conference — Submitted to ICLR 2024_

### Official Review · Reviewer_sZry · 2023-10-13

**Soundness:** 2 fair
**Presentation:** 2 fair
**Contribution:** 2 fair
**Rating:** 5
**Confidence:** 5

**Summary:**

This paper proposes a new method for training multiple student models simultaneously, called meta-collaboration, which improves student accuracy for all students and beats state-of-the-art distillation baselines. The method allows for flexible management of inference costs at test time and can be applied to various datasets and model architectures. The authors demonstrate the effectiveness of their approach on CIFAR100 and TinyImageNet datasets and show that it outperforms other distillation methods.

**Strengths:**

1. This paper presents a relatively comprehensive experiment, including multiple datasets and many teacher-student pairs.

2. This paper is well-written and easy to understand, with clear explanations of the proposed method and experimental results.

**Weaknesses:**

While the idea of meta-learning for multi-teacher KD has potential, the claims require more empirical and analytical support. Broader experimentation, justification of design choices, and engagement with relevant studies would help validate the work's novelty and significance.

1. Lack of novelty:

While multi-teacher distillation is not a new idea, the paper claims to introduce a meta-learning approach to optimize the weights of different teachers. Direct comparisons to related work like AEKD (NeurIPS2020)  that uses optimization methods are missing. More discussion is needed to justify the claims of novelty. Also, meta-optimization is very difficult to implement and not very effective. Currently meta-optimized distillation is generally ineffective or difficult to reproduce

2. Lack of thorough evaluation:

The evaluation on small datasets is insufficient. Following KD work norms, it should test on large-scale datasets like ImageNet and report downstream transfer learning results. The choice of teacher-student pairs could be better motivated by discussing alternatives like lightweight networks.

3. Lack of discussion on relevant studies:

To properly situate this work in the rapidly advancing KD literature, it needs to discuss closely related recent papers [1,2,3,4,5,6] like those pointed out. These address self-supervised KD, representation matching, offline-online transfer, architecture search for distillation, and automated KD - all highly pertinent topics the paper does not engage with. A more comprehensive literature review would strengthen the paper.

[1] Self-Regulated Feature Learning via Teacher-free Feature Distillation. ECCV2022
[2] NORM: Knowledge Distillation via N-to-One Representation Matching. ICLR2023
[3] Shadow Knowledge Distillation: Bridging Offline and Online Knowledge Transfer. NIPS2022
[4] DisWOT: Student Architecture Search for Distillation WithOut Training. CVPR2023
[5] Automated Knowledge Distillation via Monte Carlo Tree Search. ICCV2023
[6] KD-Zero: Evolving Knowledge Distiller for Any Teacher-Student Pairs. NeurIPS2023

**Questions:**

See Weaknesses.

---

> ### Author Response · Authors · 2023-11-22
>
> We express our heartfelt gratitude to the reviewer for their dedicated time and thorough evaluation of our paper. The valuable insights they offered are highly appreciated, and we are grateful for the constructive feedback. We are devoted to addressing the reviewer's concerns and integrating their suggestions into the final version, with the goal of elevating the overall quality of our paper. In the following section, we present comprehensive responses to the reviewer's comments, queries, and recommendations, aiming to enhance the paper's overall rating.
>
> ```
> Lack of novelty
> ```
> Thank you for your insightful comments and for highlighting this clarity concern. We want to address a potential misunderstanding. This paper does not involve multi-teacher distillation; rather, it focuses on a single pre-trained teacher model, with all other models being student models that are concurrently undergoing training.
>
> A significant contribution of this paper is the enhancement of knowledge distillation efficacy through the implementation of instance-wise weights, particularly in scenarios where a substantial learning capacity gap exists between the teacher model and the student model. In the conventional knowledge distillation framework, the final loss is a combination of cross-entropy loss and KL-divergence-based KD loss. Our objective is to learn weights for each of these components in an instance-wise manner. This approach allows us to concentrate on hard labels for specific data points and soft labels from the teacher for another set of points, thereby achieving an optimal trade-off between learning from the teacher model and the hard labels.
>
> Additionally, beyond learning instance-wise and loss component-wise weights, this work aims to capitalize on the learning patterns of student cohorts. This is realized through the utilization of C-Net.
>
> * meta-optimization is very difficult to implement and not very effective*
> Meta-learning is often used in setting to improve a model's generalization  and denoise setting. We point to a set of relevant literature in the Related work section.
>
> ```
> Lack of thorough evaluation
> ```
> We utilized teacher-student pairs from other papers addressing significant capacity differences between teacher and student models. While we were unable to complete the additional ImageNet experiments within the review period, we expect our method would enjoy significant gains over tranditional KD even on large scale datasets.
>
> ```
> Lack of discussion on relevant studies
> ```
> Thanks for pointing out the additional literature. However, again there seems to be some misunderstanding. The primary objective of this paper is to enhance the effectiveness of knowledge distillation through the utilization of instance-wise weights. In scenarios characterized by a substantial learning capacity disparity between the teacher model and the student model, knowledge distillation efficacy is frequently suboptimal and, at times, inferior to learning with hard labels. The papers referenced seem to improve distillation by introducing additional information or modifying the student model itself. Nevertheless, we will conduct a thorough comparison and contrast of our work with these referenced papers.

---

### Official Review · Reviewer_uMpm · 2023-10-24

**Soundness:** 3 good
**Presentation:** 3 good
**Contribution:** 2 fair
**Rating:** 3
**Confidence:** 4

**Summary:**

The paper proposes to adopt multiple students with different model architectures and a single pre-trained teacher. \
To fully utilize the multiple students, they devise C-net and consensus regularization.

**Strengths:**

1. The motivation is sound.
- models with different architecture learn different knowledge, and therefore, could be beneficial for KD.

2. The proposed C-Net and its training process are novel.
- meta-learning with bilevel optimization is straightforward.

**Weaknesses:**

1. Critical missing related work on bidirectional knowledge distillation with multiple models. \
[A]  Deep mutual learning. CVPR'18 \
[B]  Dual learning for machine translation. NeurIPS'16 \
[C]  Bidirectional Distillation for Top-K Recommender System, WWW'21 \
and on consensus learning with heterogeneous models.  \
[D] Consensus Learning from Heterogeneous Objectives for One-Class Collaborative Filtering, WWW'22

These existing works, especially [A] and [C] should be compared theoretically and empirically in the manuscript.

**Questions:**

1. Please refer to Weaknesses

2. In Eq.7, in my opinion, the optimization of $\phi$ should be the outer loop, since Eq.7 is the optimization for C-Net

---

> ### Author Response · Authors · 2023-11-22
>
> We express our sincere gratitude to the reviewer for investing their time and energy in reviewing our paper. The invaluable and insightful feedback they offered is highly appreciated, and we are grateful for the constructive comments. We are dedicated to addressing the reviewer's concerns and integrating their suggestions into the final version, with the goal of improving the overall quality of our paper. In the following section, we offer comprehensive responses to the reviewer's comments, questions, and suggestions, with the aim of enhancing the paper's evaluation.
>
> We present comparison with the additional baselines requested. We found “[C] Bidirectional Distillation for Top-K Recommender System, WWW'21 and on consensus learning with heterogeneous models.” to be specific to recommender systems and therefore compare against [1], another Bidirectional knowledge distillation method with is similar to our setup.
>
> [1] Li, Lujun, and Zhe Jin. "Shadow knowledge distillation: Bridging offline and online knowledge transfer." Advances in Neural Information Processing Systems 35 (2022): 635-649.
>
>
> | Teacher                 | Student            | CE   | KD   | DML[A]  | SKD[1]  | Ours |
> |--------------------------|---------------------|---------|------|-------------|------------|--------|
> || Resnet10_xxs | 31.85| 33.45| 33.41     | 31.59    | 36.19|
> | Resnet10_l ( 72.2) | Resnet10_xs  | 42.75| 44.87| 42.16      | 38.53 | 47.57 |
> |                                | Resnet10_s   | 52.48  | 55.38| 51.13| 48.11| 58.36|
> || Resnet10_m   | 64.28          | 66.93| 62.06| 56.97| 69.42|
> |--------------------------|---------------------|---------|------|-------------|------------|--------|
> || Resnet10_xxs | 31.85          | 33.95| 32.04| 31.03| 35.97|
> | Resnet34  (79.47)   | Resnet10_xs | 42.75| 44.87| 42.47 | 39.56 | 47.53|
> || Resnet10_s   | 52.48          | 55.56| 51.22| 47.19| 58.1 |
> | |Resnet10_m   | 64.28          | 67.27| 63.12| 56.8 | 69.21|
>
> ```
> In Eq.7, in my opinion, the optimization of \phi should be the outer loop, since Eq.7 is the optimization for C-Net.
> ```
> Due to the bi-level optimization formulation, the C-Net parameters undergo updates based on the most recent model parameters. Initially, we adjust the model (student) parameters using loss gradients derived from the weighting parameters acquired from C-Net. Consequently, the updated parameters depend on the weights obtained from C-Net. Subsequently, employing these updated parameter values, we calculate loss gradients for C-Net.

---

### Official Review · Reviewer_cbio · 2023-10-31

**Soundness:** 2 fair
**Presentation:** 2 fair
**Contribution:** 2 fair
**Rating:** 5
**Confidence:** 4

**Summary:**

The paper proposes a distillation method called meta-collaboration, where multiple student models of different capacities are simultaneously distilled from a single teacher model. The students improve each other through information sharing during distillation by a C-Net module. The method outperforms compared distillation baselines.

**Strengths:**

1) The proposed method is evaluated on a wide range of student and teacher architectures, as well as model sizes.
2) The paper is well-written and effectively communicates the key ideas, methodology, and experimental results. The organization of the paper is logical.
3) The paper provides detailed implementation details and code, which is easy to reproduce.

**Weaknesses:**

1) The novelty is limited in my view. This work follows the widely used online distillation framework, except that the proposed C-NET part is trained with meta-learning. There are no clear improvements to the framework.
2) The experiments are only conducted on small datasets (CIFAR100, TinyImageNet). The author is encouraged to evaluate your method on a large dataset like ImageNet-1K.
3) Limited evaluation and ablation studies on modern high-performance CNN architectures such as ConvNext[1], VAN[2], and RepLKNet[3].
4) Can you provide further theoretical analysis or insights into how the meta-collaboration process influences the learning patterns of the student models?


[1]  Woo et al. Convnext v2: Co-designing and scaling convnets with masked autoencoders. Arxiv, 2023.
[2] Guo et al. Visual Attention Network. CVMJ 22.
[3] Ding et al. Scaling up your kernels to 31x31: Revisiting large kernel design in cnns. CVPR 22.

**Questions:**

Please refer to weaknesses.

---

> ### Author Response · Authors · 2023-11-22
>
> We express our genuine gratitude to the reviewer for investing their time and energy in reviewing our paper. The valuable feedback offered is highly appreciated, and we are grateful for the constructive comments. We are dedicated to addressing the concerns raised by the reviewer and integrating their suggestions into the final version to improve the overall quality of our paper. In the following section, we offer thorough responses to the reviewer's comments, questions, and suggestions, aiming to elevate the rating of our paper.
>
> ```
> The novelty is limited in my view. This work follows the widely used online distillation framework, except that the proposed C-NET part is trained with meta-learning. There are no clear improvements to the framework.
> ```
>
> A primary contribution of this paper lies in enhancing the effectiveness of knowledge distillation by incorporating instance-wise weights in situations characterized by a significant disparity in learning capacity between the teacher and student models. In the typical framework of any knowledge distillation, the final loss is a combination of cross-entropy loss and KL-divergence-based KD loss. We aim to learn weights for each of these components on an instance-wise basis. This approach allows us to emphasize hard labels for specific data points and soft labels from the teacher for another set of points, facilitating an optimal trade-off between learning from the teacher model and the hard labels. This trade-off is crucial as diverse students exhibit varying learning capacities, and when a student model is considerably smaller than the teacher model, some learning from hard labels proves beneficial.
>
> Beyond learning instance-wise and loss component-wise weights, our work seeks to leverage insights from student cohorts, inspired by online distillation. However, unlike traditional online distillation, there is no explicit matching imposed among the student models through a loss function. Instead, this information is implicitly shared among the student models through the utilization of a common C-Net.
>
> ```
> Can you provide further theoretical analysis or insights into how the meta-collaboration process influences the learning patterns of the student models?
> ```
>
> The effect on learning patterns of different student models while training with weights from C-Net like neural network is well studied in  [1]. The same setting is applicable here as well.
>
> [1]Shu, Jun, et al. "Meta-weight-net: Learning an explicit mapping for sample weighting." Advances in neural information processing systems 32 (2019).

---

### Official Review · Reviewer_Bq77 · 2023-11-02

**Soundness:** 3 good
**Presentation:** 3 good
**Contribution:** 2 fair
**Rating:** 5
**Confidence:** 4

**Summary:**

This paper proposes a framework for distilling the knowledge from a pretrained teacher to multiple students simultaneously. It uses a trainable network to learn the cross-entropy and KL-divergence weights for the K students. This trainable network is trained with a bi-level optimization strategy. The K students are also supervised by the PooledStudent logits. Experiments show promising results.

**Strengths:**

1. This paper is clearly written.
2. Distilling a pertained teacher to multiple students simultaneously is an interesting question.

**Weaknesses:**

1. The novelty looks incremental. It looks to me that the paper combines offline KD and online KD by training multiple students simultaneously while learning the loss weights with the meta-learning strategy (bi-level optimization).
2. Only two small datasets CIFAR-100 and tiny-ImageNet are used.
3. More comprehensive ablation studies about the weight-learning strategy (e.g., C-NET) and the PooledStudent logits should be conducted.

**Questions:**

1. It misses some ablation studies to show how the learned weights by C-NET are better than other weight strategies, e.g., manually set.
2. Why use PooledStudent logits as the supervision? Theoretical or empirical explanation is supposed to be provided.
3. Experiments on large-scale datasets, e.g., ImageNet, should be reported.

---

> ### Author Response · Authors · 2023-11-22
>
> We extend our sincere appreciation for the reviewer's dedicated time and effort in evaluating our paper. The insightful feedback provided is invaluable to us, and we are thankful for the constructive comments. We are committed to addressing the reviewer's concerns and incorporating their suggestions in the final version, aiming to enhance the overall quality of our paper. Below, we provide detailed responses to the reviewer's comments, questions, and suggestions with the hope of improving the rating of our paper.
>
> ```
> - The novelty looks incremental. It looks to me that the paper combines offline KD and online KD by training multiple students simultaneously while learning the loss weights with the meta-learning strategy (bi-level optimization).
> - It misses some ablation studies to show how the learned weights by C-NET are better than other weight strategies, e.g., manually set.
> ```
>
> Thank you for your insightful comments and for bringing attention to the clarity issue. We wish to address a potential misunderstanding. A key contribution of this paper is the enhancement of knowledge distillation efficacy through the incorporation of instance-wise weights. In scenarios where there exists a significant learning capacity gap between the teacher model and the student model, knowledge distillation efficacy is often suboptimal, and in some cases, inferior to learning with hard labels.
>
> To alleviate this issue, our proposal involves the utilization of instance-wise and loss component-wise weights. Typically, in any knowledge distillation framework, the final loss is a combination of cross-entropy loss and KL-divergence based knowledge distillation (KD) loss. Our objective is to learn weights for each of these components on an instance-wise basis. This approach enables us to focus on hard labels for specific sets of data points and soft labels from the teacher for another set of points. This allows us to achieve an optimal trade-off between learning from the teacher model and the hard labels, a crucial consideration given the diverse learning capacities of different students. When a student model is significantly smaller than the teacher model, some learning from hard labels proves to be beneficial.  In addition to learning instance-wise and loss component-wise weights in this work we also want to benefit from learning patterns of the student cohorts. This is achieved via using C-Net.
>
> ```
> Why use PooledStudent logits as the supervision? Theoretical or empirical explanation is supposed to be provided.
> ```
>
> In the context of handling a knowledge distillation scenario characterized by a substantial disparity between the student and teacher models, it has been observed that incorporating outputs from intermediate models—those with learning capacities between the teacher and student models—enhances the knowledge distillation process. Consequently, we introduce the PooledStudent logits in conjunction with the teacher logits to amplify the knowledge transfer from the teacher model to the student model.  Here again, we control the participation of the PooledStudent logits in the training process using instance-wise loss-component-wise loss.  Further in Appendix B.1 we study the effect of the addition of PooledStudent logits.

---

### Meta-Review · Area_Chair_5G1Z · 2023-12-06

**Metareview:**

None of the reviewers is in favor of accepting the paper particularly. They argue that the methodological contribution is relatively minor and the experimental evaluation is not sufficiently thorough/convincing.

**Justification For Why Not Higher Score:**

None of the reviewers is positive about the paper.

**Justification For Why Not Lower Score:**

N/A

---

### Decision · Program_Chairs · 2024-01-16

Reject